# GOGH: Correlation-Guided Orchestration of GPUs in Heterogeneous Clusters

Ahmad Raeisi[1,5], Mahdi Dolati[2], Sina Darabi[3], Sadegh Talebi[*5], Patrick Eugster[3], and Ahmad Khonsari[1]

[1]University of Tehran
[2]Sharif University of Technology
[3]Università della Svizzera italiana (USI)
[5]University of Copenhagen
 ahmad.raeisi@di.ku.dk, mdolati@sharif.edu, sina.darabi@usi.ch, sadegh.talebi@di.ku.dk,
patrick.eugster@usi.ch, khonsari@ut.ac.ir

## Abstract

In heterogeneous clusters with varying capabilities and energy efficiency, sustainable use of mixed-generation resources is essential. We propose a method for adaptive management of machine learning jobs, aiming to minimize energy while meeting performance targets which uses two neural networks to cope with hardware utilization uncertainties. We demonstrate the efficacy of this adaptive process via the Gavel benchmark [1].

## 1 Introduction

Recent advances in machine learning have led to large models and datasets, creating high computational demands. While specialized accelerators address these needs, upgrading all infrastructure is impractical; thus, heterogeneous clusters combining legacy and modern GPUs remain common [2, 3]. Efficient scheduling is challenging due to diverse job structures, resource contention, and limited understanding of job–hardware interactions [4, 5]. Historical performance data, however, can reveal patterns useful for predicting throughput across jobs and devices [6]. We present *GOGH*, a framework for scheduling deep learning jobs on heterogeneous GPU clusters by leveraging correlations across jobs and accelerators to predict throughput and guide allocations. Its key components include: (i) a neural predictor for initial and refined throughput estimates (using RNN, Feedforward (FF), and Transformer models), and (ii) an ILP-based optimizer that assigns jobs to GPUs while meeting throughput guarantees and minimizing power use. Using the Gavel dataset [1], we demonstrate that GOGH improves both prediction accuracy and scheduling efficiency over baselines.

We make the following contributions: (1) A correlation-guided framework for heterogeneous GPU scheduling; (2) an ILP formulation balancing throughput, efficiency, and guarantees; (3) evaluation of neural architectures for prediction accuracy; and (4) experimental validation showing significant scheduling and prediction gains. We refer to the full version of the article [7] for detailed discussions.

**Related work. Heterogeneity-aware scheduling.** Gavel [1] introduced effective throughput for accelerator-aware scheduling; Pollux [8] co-adapts resource allocation and training configurations. **Resource sharing.** TGS [9] enables transparent GPU sharing at the OS layer; HiveD [10] provides virtual clusters for multi-tenant fairness and affinity. **Elastic/serverless training.** ElasticFlow [11] shows benefits of elasticity for deadline-aware training. **LLM serving.** Parrot [12] exposes semantic signals to reduce redundancy during inference. Our work differs by learning correlations to predict cross-devicethroughput and refining estimates online to guide energy-aware ILP-based allocation.

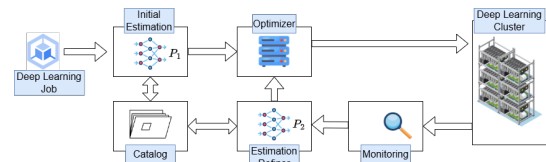

**Figure 1.** Architecture of GOGH.

## 2 GOGH Design

GOGH is a solution that dynamically decides GPU–job assignments in a deep learning cluster, as ML jobs arrive over time. Figure 1 illustrates the architecture of GOGH.

At its core, GOGH has an ILP-based *Optimizer* module that assigns jobs to GPUs subject to capacity and minimum-throughput constraints while minimizing power. However, job throughput on each GPU type is not known by the *Optimizer* apriori. Thus, GOGH uses four helper modules to improve its assignment in time. $P_1$ provides initial per-(job, accelerator, co-location) throughput estimates from a continuously updated *Catalog* of historical runs.

---

*Corresponding Author.

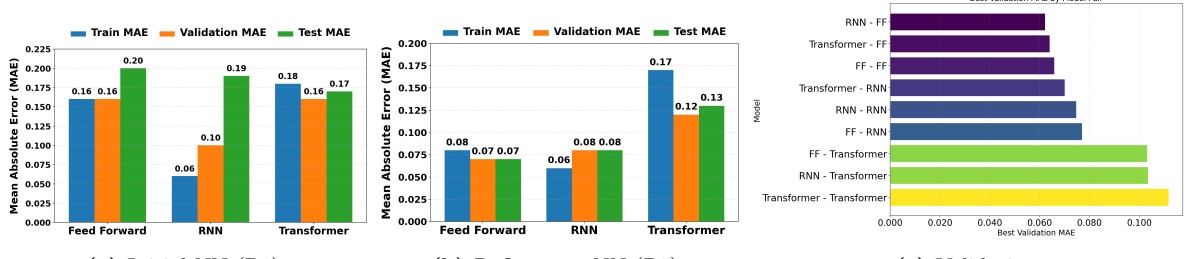

**(a)** Initial NN (P1).     **(b)** Refinement NN (P2).     **(c)** Validation

**Figure 2.** MAE of models; Combined validation of P1–P2 model pairs.

*Monitor* records realized throughput; $P_2$ refines the Catalog using these observations, improving future allocations. To cast this as an optimization problem, let $\mathcal{S}$ be servers, $\mathcal{A}$ GPU types, and $\mathcal{J}$ ML jobs. GPU capacity of type $a$ is $K_a$, which in practice for most GPUs is one or two. Define $\mathcal{C} \subseteq \{c \subseteq \mathcal{J} : |c| \leq 2\}$ as job combinations (job co-location) that can be assigned to GPUs. Let $T_{a,j}^c$ be throughput of job $j$ on type $a$ under co-location $c$. As is unknown, we define $\widetilde{T}_{a,j}^{i,c}$ as the i-th estimate of $T_{a,j}^c$ computed by $P_1$ and $P_2$. Specifically, $P_1$ writes the initial estimate $\widetilde{T}_{a,j}^{0,c}$ into the Catalog and $P_2$ refines it after monitoring round $i > 0$.

Given estimated or measured throughput values, we allocate GPUs to jobs via an integer linear program (ILP) that minimizes total power consumption while meeting throughput and capacity constraints. We define binary variables $x_{a,s}^c$ indicating whether job combination $c$ is assigned to accelerator $a$ on server $s$. The objective (1a) minimizes power $\gamma_a(x)$, obtained from profiling. Constraints (1b)–(1e) ensure that each job is scheduled, accelerator capacities are respected, and minimum throughput $T_j$ is met. We solve the formulation with a standard ILP solver; more efficient algorithms are left for future work.

$$\min_{x_{a,s}^c \in \{0,1\}} \sum_{a \in \mathcal{A}} \gamma_a \left( \sum_{c \in \mathcal{C}_j} T_{a,j}^c x_{a,s}^c \right) \quad (1a)$$

$$\text{s.t. } 1 \leq \sum_{s,a,c \in \mathcal{C}_j} x_{a,s}^c \leq D_j, \quad (1b)$$

$$\sum_{c \in \mathcal{C}} |c|\, x_{a,s}^c \leq K_a, \quad (1c)$$

$$\overline{T}_j \leq \sum_{a,c \in \mathcal{C}_j} T_{a,j}^c x_{a,s}^c, \quad (1d)$$

$$\sum_{c \in \mathcal{C}} x_{a,s}^c \leq 1. \quad (1e)$$

## 3   Evaluation

We use the Gavel benchmark [1], reporting throughput (iterations/sec) for diverse ML workloads (listed in Table 1) on six GPU types (k80, p100, v100 and unconsolidated variants). As for models, we implemented FF, RNN, and Transformer variants

**Table 1.** Workloads in simulation.

| Application | Batch Sizes |
|---|---|
| ResNet-18 | $\{16, 32, 64, 128, 256\}$ |
| ResNet-50 | $\{16, 32, 64, 128, 256\}$ |
| Transformer | $\{16, 32, 128, 256\}$ |
| LM | $\{5, 10, 20, 80\}$ |

for both $P_1$ and $P_2$, with similar complexity. Models were evaluated using the mean absolute error (MAE) across the training, validation, and test sets.

**Results.** *Initial Estimation ($P_1$).* On validation, RNN achieved the lowest MAE, while on test the Transformer generalized best. RNN fit training data strongly, but Transformer was more robust to distribution shift. *Refinement ($P_2$).* FF provided the most consistent refinement with the lowest validation and test MAE, outperforming RNN and Transformer despite RNN's lower training loss. Pairing $P_1$ and $P_2$ shows that *RNN→FF* yields the best overall validation MAE, outperforming *Transformer→FF* by $\approx 2.8\%$ in our setting. Capturing temporal patterns initially (RNN) and refining deterministically (FF) provides strong accuracy and generalization; see [7] for a detailed discussion.

**Discussion.** Estimates are stable under batch-size perturbations and reflect co-scheduling contention: similar jobs map to accelerators with comparable predicted throughput; conflicting co-locations show predicted degradation aligned with observations. These behaviors indicate that GOGH captures structural properties of workload–device interactions without hand-tuned rules.

## 4   Conclusion

We propose GOGH, a correlation-guided framework for managing GPUs in heterogeneous clusters. GOGH estimates job throughput across GPU types by leveraging inter-job and inter-GPU correlations, refining predictions with runtime observations. An ILP allocator minimizes power while meeting throughput goals. Among tested architectures, the RNN–FF pipeline achieved prediction errors as low as 5%. Future work includes ILP approximations and reinforcement learning for adaptive scheduling.

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
