# OpenReview forum: "GOGH: Correlation-Guided Orchestration of GPUs in Heterogeneous Clusters"
_NLDL.org/2026/Abstracts_Track — NLDL 2026 Abstracts_

### Official Review · Reviewer_8YGd · 2025-11-02

**Soundness:** 3
**Correctness:** 3
**Rating:** 4
**Confidence:** 3

**Summary:**

The abstract proposes a method for managing gpu jobs based on using two neural networks and evaluate the approach on the Gavel benchmark.

**Strengths:**

- The problem of managing gpu jobs is significant, and the idea of using deep learning models for this purpose is interesting.
- The description of the model architecture is well written and figure 1 clearly shows the overall framework.
- The evaluation is thorough testing across different GPU types and models

**Weaknesses:**

- Including non-deep learning approaches in the evaluation would provide a more comprehensive evaluation of the approach in comparison to other methods.

---

### Official Review · Reviewer_z6o5 · 2025-11-03

**Soundness:** 3
**Correctness:** 3
**Rating:** 5
**Confidence:** 4

**Summary:**

This extended abstract is about the efficient scheduling of NN-based ML tasks on GPU clusters. The proposed approach *correlation-**G**uided **O**rchestration of **G**PUs in **H**eterogeneous clusters* consists of two neural networks and an integer linear program-based optimizer. The first NN exploits correlations between tasks and predicts initial throughput estimates while the second NN refines those estimates. Their proposed approach is then evaluated on the Gavel benchmark and preliminary results are presented.

Apparently, a full version of this extended abstract is available [4].

**Strengths:**

- Efficient scheduling of ML jobs on large GPU clusters is of interest to the community.
- The energy minimizing aspect of the approach is appealing and is important nowadays.
- The initial experimental results are promising.

**Weaknesses:**

- Parts of the paper could be written in a more clear and accessible way. See the minor comments below.
- From the experimental results, it is unclear if the proposed approach is indeed more time- and energy-efficient than, e.g., a random scheduler.
- It is unclear what the overhead if GOGH is.
- Line 035 mentions baselines but I do not see any comparison to any other scheduler.
- It would be interesting to see how and on what the correlations are computed. If I design a NN-based ML task, what information is used by GOGH to schedule it?


Minor comments:
- Line 68 mentions **four** helper modules. What are the helper modules? $P_1$ and $P_2$? But where are $P_3$ and $P_4$ then? Lines 111-115 define $P_1$ and $P_2$ a bit better.
- What is $D_j$?
- Where does the initial catalogue come from?
- What exactly is meant by $i$-th estimate?
- It is unclear what train, validation, and test refer to. Are those the sets of Gavel?
- What about tracking carbon emissions using, e.g., using carbontracker (arXiv:2007.03051)

---

### Decision · Program_Chairs · 2025-11-05

**Decision:**

Accept

**Comment:**

The abstract is of interest to the community and should be presented at the conference.